# Ca$^{2+}$-Daptomycin targets cell wall biosynthesis by forming a tripartite complex with undecaprenyl-coupled intermediates and membrane lipids

Fabian Grein [1,2,5], Anna Müller [1,5], Katharina M. Scherer [3,5], Xinliang Liu[3], Kevin C. Ludwig[1], Anna Klöckner[1,2], Manuel Strach[1], Hans-Georg Sahl[4], Ulrich Kubitscheck [3✉] & Tanja Schneider[1,2✉]

The lipopeptide daptomycin is used as an antibiotic to treat severe infections with gram-positive pathogens, such as methicillin resistant *Staphylococcus aureus* (MRSA) and drug-resistant enterococci. Its precise mechanism of action is incompletely understood, and a specific molecular target has not been identified. Here we show that Ca$^{2+}$-daptomycin specifically interacts with undecaprenyl-coupled cell envelope precursors in the presence of the anionic phospholipid phosphatidylglycerol, forming a tripartite complex. We use micro-biological and biochemical assays, in combination with fluorescence and optical sectioning microscopy of intact staphylococcal cells and model membrane systems. Binding primarily occurs at the staphylococcal septum and interrupts cell wall biosynthesis. This is followed by delocalisation of components of the peptidoglycan biosynthesis machinery and massive membrane rearrangements, which may account for the pleiotropic cellular events previously reported. The identification of carrier-bound cell wall precursors as specific targets explains the specificity of daptomycin for bacterial cells. Our work reconciles apparently inconsistent previous results, and supports a concise model for the mode of action of daptomycin.

[1] Institute for Pharmaceutical Microbiology, University Hospital Bonn, University of Bonn, Bonn, Germany. [2] German Center for Infection Research (DZIF), partner site Bonn-Cologne, Bonn, Germany. [3] Institute for Physical and Theoretical Chemistry, University of Bonn, Bonn, Germany. [4] Institute of Medical Microbiology, Immunology and Parasitology, University Hospital Bonn, University of Bonn, Bonn, Germany. [5] These authors contributed equally: Fabian Grein, Anna Müller, Katharina M. Scherer. ✉email: u.kubitscheck@uni-bonn.de; tschneider@uni-bonn.de

  

D aptomycin (DAP) is an antibiotic with unprecedented biophysical properties and antibacterial activities. Being initially considered unsuitable for clinical application due to toxic myopathies, a revised once-daily administration scheme was approved by the US Food and Drug Administration (FDA) in 2003 for the treatment of complicated skin and soft-tissue infections caused by various Gram-positive pathogens, making it the first lipopeptide antibiotic in clinical use.

DAP (previously designated LY146032), produced by the soil bacterium *Streptomyces roseosporus*, is a depsipeptide with a 10-membered cyclic lactone core that contains a number of unusual non-proteinogenic and D-amino acids cyclized by an ester bond. Three exocyclic amino acids (Trp1, Asn2, Asp3) link the decapeptide core to a decanoyl fatty acid side chain[1,2]. Structurally, DAP resembles a group of acidic lipopeptides, including A54145, friulimicin, tsushimycin and amphomycin[3,4]. Apart from structural features these peptides share a strict requirement of $Ca^{2+}$ ions for antimicrobial activity. Importantly, the complex formation with $Ca^{2+}$ affects the physicochemical properties of DAP, masking the anionic nature and conferring an overall amphiphilic character. Since this is a typical feature of cationic antimicrobial peptides (cAMPs) it was suggested that DAP likewise would act as cAMPs, which primarily target negatively charged microbial membranes[5]. In line with an AMP-like mode of action, mechanisms of DAP resistance overlap with those observed for cAMPs, e.g. modulation of cell surface charge and membrane characteristics[6], suggesting that the main target of DAP is or is located within the cytoplasmic membrane. However, and despite considerable experimental studies, the mechanism of action is still not completely understood. In particular, a molecular target had not been identified so far.

DAP mechanism of action studies produced controversial results for several decades. The earliest reports suggested that DAP inhibits peptidoglycan biosynthesis, accompanied by potassium leakage from *S. aureus* cells[7]. Later, cell division and synthesis of secondary cell wall polymers (i.e. lipoteichoic acid) had been proposed as target sites of DAP action[8,9]. Diverse membrane perturbing mechanisms have been accounted for the bactericidal effect of DAP, including induction of altered membrane curvature, membrane depolarisation and pore formation (e.g. Silverman et al.[10]). However, based on analysis of killing kinetics, it was also suggested that membrane leakage would be the result rather than the cause of cell death[11]. More recently, DAP was shown to interfere with fluid lipid microdomains in the membranes of susceptible bacteria, resulting in a drastic rearrangement of local membrane architecture, followed by the delocalisation of essential peripheral membrane proteins, such as the lipid II synthase MurG[12].

From various experiments, two currently prevailing hypotheses for the mechanism of action of DAP emerged. One model, originating from structural studies and the observed membrane depolarising effect, suggests that $Ca^{2+}$-DAP forms oligomeric aggregates, which upon contact with phosphatidylglycerol (PG) rearrange into a pore-like complex, leading to ion leakage and dissipation of membrane potential[5]. Generally, the presence of PG in membranes was shown in various studies to be a prerequisite for DAP activity (e.g. Jung et al. and Muraih et al.[11,13]). The second model suggests that DAP insertion at specific membrane domains, also enriched in PG, affects the physicochemical properties of the cytoplasmic membrane, triggering pleiotropic effects on essential cell wall biosynthesis and cell division processes[9,12].

Here, we report the identification of specific DAP targets, which allows to amalgamate controversially discussed results of previous mechanism of action studies. Using comprehensive in vivo and in vitro approaches in combination with fluorescence and optical sectioning microscopy, we show that DAP specifically interacts with undecaprenyl-coupled cell envelope precursors in the presence of PG by forming a tripartite complex.

## Results

**DAP interferes with the lipid II biosynthesis cycle.** Previous studies described various DAP-mediated effects using different model organisms and various experimental approaches, which resulted in diverse mechanism of action models for DAP[5,10,12,14]. The differences in experimental approaches and conditions largely hampered direct and unrestricted data comparison. To consolidate previous findings with in vivo and in vitro approaches conducted in this study, we revised selected key experiments under consistent conditions.

Early investigations on the mechanism of action of DAP proposed peptidoglycan biosynthesis as the primary target pathway of the lipopeptide antibiotic[7,15]. Corroborating, macromolecular incorporation assays revealed a preferential inhibition of the incorporation of glucosamine into cell wall, while other major biosynthesis pathways remained almost unaffected[12].

DAP had previously been demonstrated to elicit the LiaRS stress response, supporting a specific inhibition of cell wall biosynthesis. The LiaRS two-component system (TCS) is known to respond to antibiotics that interfere with the lipid II biosynthesis cycle[16,17].

In search for a molecular target within the peptidoglycan biosynthesis pathway we here revisited the effect of DAP on the LiaRS response, but monitored $P_{liaI}$-*lux* induction over time and tested the impact of $Ca^{2+}$. Evaluating bioluminescence revealed a concentration-dependent induction by DAP (Fig. 1a, Supplementary Fig. 1) similar to teixobactin, which targets bactoprenyl-bound peptidoglycan precursors[18]. DAP promoted *lux* expression was only observed in the presence of 1.25 mM $Ca^{2+}$, indicating response specificity and suggesting that DAP impacts on the membrane-bound peptidoglycan biosynthesis machinery. The peptidoglycan biosynthesis machinery is predominantly localised at the division septum in staphylococci[19]. Accordingly, we found Bodipy-FL-labelled DAP to preferentially localise to this cellular site in *S. aureus* (Fig. 1b). Results from this experiment were validated by fluorescence microscopy of native DAP, taking advantage of the naturally fluorescent amino acid kynurenine, to rule out any disturbing effect of the fluorescence label (Fig. 1c).

Monitoring DAP binding over time showed, that binding is dynamic and occurs in two phases (Fig. 2). In phase I (0.5–15 min after DAP exposure) DAP septal localisation is prevailing. Prolonged incubation resulted in a progressive dispersion of DAP throughout the entire cytoplasmic membrane (phase II), finally resulting in membrane collapse and cell shrinkage (Fig. 2a). Importantly, initial septal DAP binding in phase I appears to correlate with killing (Fig.2b), indicating that cell death is the consequence of septal binding. In agreement, a subset of the cells in phase I were found to be dead as determined by sytox green staining (Supplementary Fig. 2).

To study the interaction of DAP with putative target molecules within the peptidoglycan biosynthesis pathway (Supplementary Fig. 3), we analysed the effect of various cell wall targeting antibiotics on the binding of DAP-FL to *S. aureus* cells. Short-term pre-incubation (2 min) of cells with antibiotics that bind to cell wall lipid intermediates, such as the ultimate peptidoglycan precursor lipid II, strongly diminished DAP binding, suggesting a blockage of target access (Fig. 3a, b). Both, teixobactin and oritavancin, known to interact with bactoprenyl-coupled cell wall intermediates, reduced DAP binding up to 50%, while pretreatment with the β-lactam oxacillin did not have any detectable effect. Of note, pre-incubation with teixobactin partially protected

*S. aureus* from DAP and significantly delayed DAP-induced killing, indicating that both compounds interact with the same target (Fig. 3c).

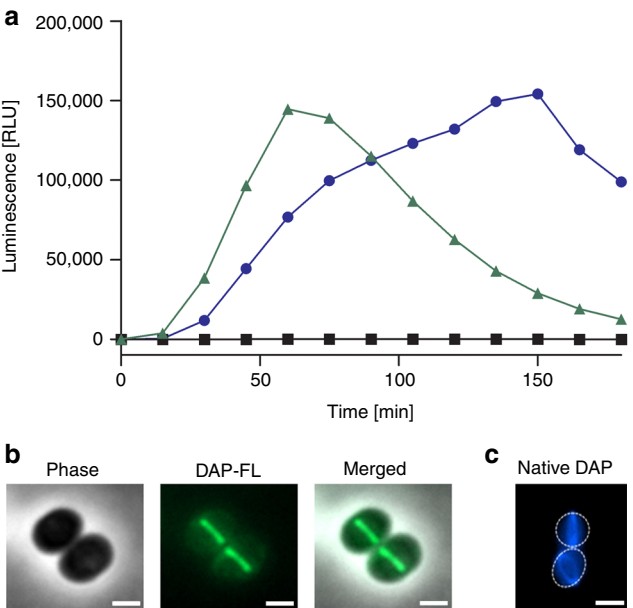

**Fig. 1 DAP specifically induces the LiaRS stress response in the presence of $Ca^{2+}$ and binds to the cell division site. a** $P_{liaI}$-*lux* induction in *B. subtilis* was monitored upon addition of DAP (0.5 µg ml$^{-1}$) in the absence (black line) or presence (blue line) of 1.25 mM $Ca^{2+}$ and teixobactin (0.5 µg ml$^{-1}$) (green line). Luciferase activity is presented as relative luminescence units (RLU). Representative graph of three independent experiments. **b, c** DAP localises to the septum of *S. aureus*. Cells were grown to mid-exponential phase (OD$_{600}$ = 0.5) followed by the addition of $Ca^{2+}$ and a mixture of labelled and unlabelled DAP (7 µg ml$^{-1}$ DAP; 0.8 µg ml$^{-1}$ DAP-FL) **b** or unlabelled DAP (7 µg ml$^{-1}$) **c**. Cells were washed and imaged by fluorescence microscopy; phase, phase contrast. Cell outlines in **c** are indicated by dashed lines. Scale bar: 1 µm. Representative images from five independent experiments are shown. Source data are provided as a Source Data file.

The cell wall precursor lipid II is synthesised on the cytoplasmic site of the bacterial membrane by a cascade of enzymatic reactions (Supplementary Fig. 3) before the bactoprenyl phosphate-linked disaccharide-pentapeptide is translocated to the outer leaflet, where lipid II is readily accessible to even large antibiotics[20]. Apart from its role as an essential cell wall building block, lipid II plays crucial roles as a functional scaffold coordinating cell wall biosynthesis and cell division. In this context, lipid II is discussed to contribute to the recruitment and assembly of these protein complexes[20–22].

To investigate the cellular effects triggered by DAP septum binding, we examined the co-localisation of DAP with the putative lipid II flippase FtsW. To this end *S. aureus* expressing a FtsW-GFP fusion protein was incubated with fluorescently labelled DAP (DAP-TMR) for 0.5 min (phase I). As revealed by dual-colour fluorescence microscopy using highly inclined and laminated optical (HILO) sheet illumination, FtsW-GFP and DAP-TMR co-localise to the division septum (Supplementary Fig. 4a, Supplementary Movie 1). In line with the observed dispersed distribution of DAP after prolonged incubation (20 min, phase II), FtsW-GFP was similarly delocalised from the division septum and found to accumulate in membrane spots together with DAP (Supplementary Fig. 4), further supporting a specific interaction with lipid II.

While the short-term exposure with lipid II-binding antibiotics blocked the interaction with the molecular target (Fig. 3a, b), we took advantage of the fact that the prolonged treatment of cells with sublethal concentrations of vancomycin results in an accumulation of lipid II[23]. Pre-incubation with vancomycin for 30 min resulted in strongly elevated cellular lipid II levels triggering hyperaccumulation of DAP (Supplementary Fig. 5a–c).

In contrast, no such hyperaccumulation was observed in untreated or oxacillin-pre-treated control cells, in which lipid II levels were unchanged (Supplementary Fig. 5c). Corroborating, DAP-mediated killing was significantly accelerated in cells with increased lipid II content (Supplementary Fig. 5d).

**DAP binding to supported bilayers doped with cell wall lipid intermediates.** In vivo experiments clearly pointed to a specific interaction of DAP with bactoprenyl-coupled cell wall precursors.

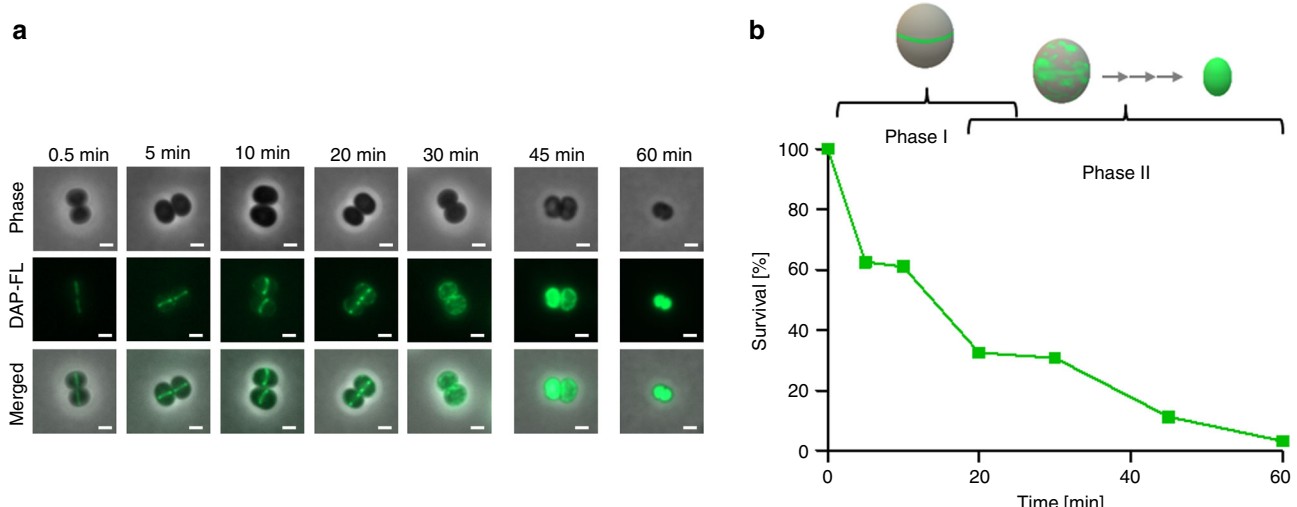

**Fig. 2 DAP binds to *S. aureus* in a biphasic manner. a** DAP-FL binding to *S. aureus* monitored over time (0–60 min). *S. aureus* HG003 was grown to mid-exponential phase (OD$_{600}$ = 0.5) followed by the addition of $Ca^{2+}$ and a mixture of labelled and unlabelled DAP (7 µg ml$^{-1}$ DAP; 0.8 µg ml$^{-1}$ DAP-FL). At different time points, samples were taken, washed and imaged by fluorescence microscopy. Representative pictures are shown from three independent experiments. Phase, phase contrast. Scale bar 1 µm. **b** Survival of cells from the experiment described in **a** and schematic depiction of DAP-FL-binding behaviour in two phases. Source data are provided as a Source Data file.

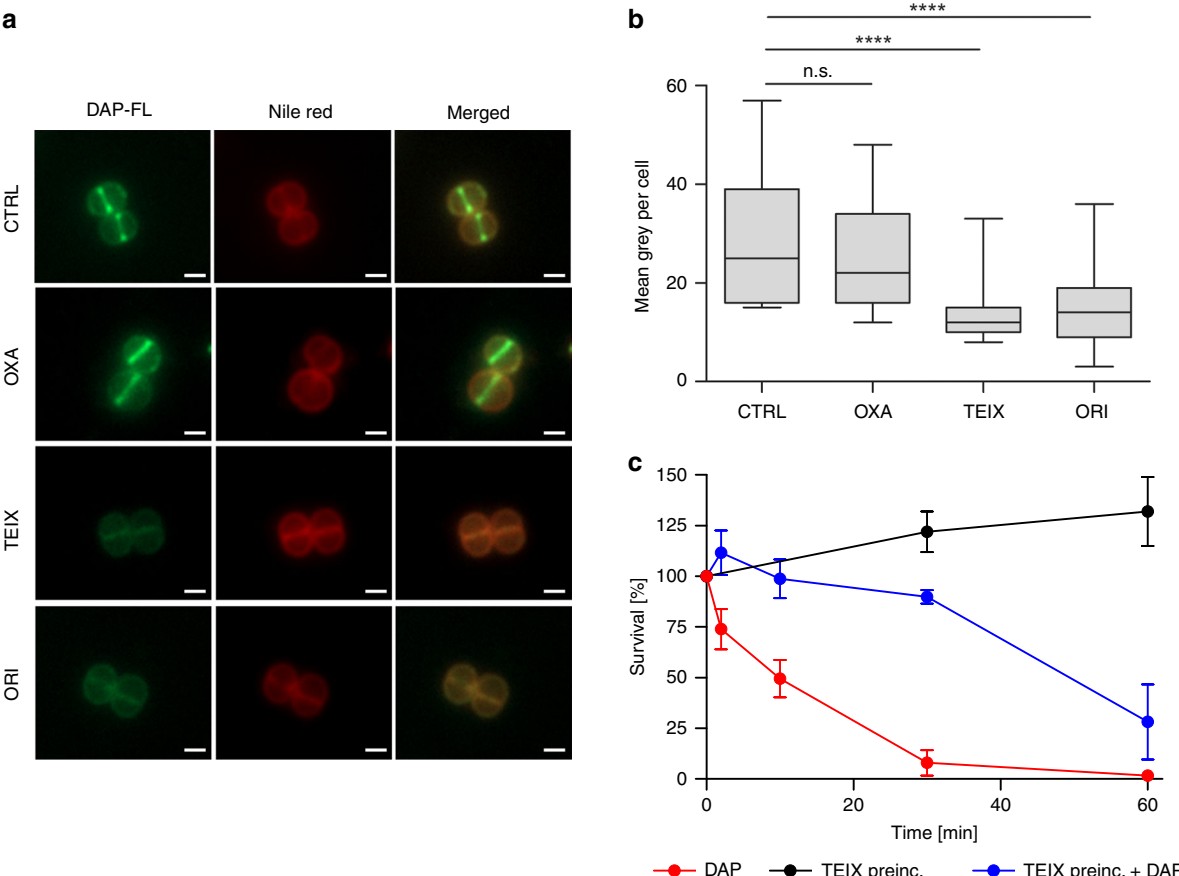

**Fig. 3 Pre-incubation with lipid II-binding antibiotics prevents DAP binding to *S. aureus* and delays DAP-induced killing. a** Binding of DAP-FL to *S. aureus* cells ($OD_{600} = 0.5$) pre-incubated with cell wall targeting antibiotics (oxacillin (OXA), teixobactin (TEIX) and oritavancin (ORI) (4-fold MIC)). After pre-incubation for 2 min, cells were washed and incubated with DAP-FL in the presence of $Ca^{2+}$ for 10 min followed by washing and the addition of nile red membrane stain. Cells were washed again and subjected to fluorescence microscopy. In a control (CTRL) pre-incubation with antibiotics was omitted. Scale bar 1 μm. Representative images are shown. **b** Quantification of DAP-FL binding measured in the experiment as described in **a**. Binding is expressed as mean grey value per cell. Box plots represent the interquartile range of the data. The black bar represents the mean and whiskers represent minimal and maximal values, respectively. At least $n = 350$ cells were evaluated for each condition from three biologically independent experiments. Significance was determined by unpaired Student's *t*-test with a 95% confidence interval. ****$p < 0.0001$, n.s., not significant ($p = 0.0835$). **c** Survival of *S. aureus* challenged with DAP (10 μg ml$^{-1}$) without (red line) or with (blue line) pre-incubation with TEIX as described in **a**. Bactericidal effects were not observed in cells only pre-treated with TEIX (black line). Data presented are mean values from $n = 3$ biologically independent experiments. Error bars represent the standard deviation (SD). Source data are provided as a Source Data file.

To test potential target molecules in a defined model system mimicking a more natural membrane environment, supported bilayers were individually loaded with $C_{55}P$, $C_{55}PP$ or lipid II (0.1 mol%) and DAP binding was monitored by total internal reflection fluorescence (TIRF) microscopy.

Basal binding of DAP to pure neutral phosphatidylcholine (PC) membranes was observed and only a moderate increase in binding was detected when bilayers were doped with individual cell wall precursors (Fig. 4a, left). In contrast, when the different bactoprenyl–lipid precursors were used in combination with PG (0.1 mol%), binding of DAP to the membrane was drastically increased (Fig. 4a, right and Supplementary Movie 2). Despite its essentiality for DAP antimicrobial activity, an equivalent increase in anionic PG (0.2 mol%) alone did not promote binding of the lipopeptide, compared to precursor containing PC-bilayers, revealing that binding does not solely rely on charge effects. As observed with whole cells, pre-incubation with antibiotics, that specifically bind peptidoglycan precursors $C_{55}P$ (friulimicin), $C_{55}PP$ (bacitracin) and lipid II (oritavancin), strongly decreased binding of DAP to the bilayer-embedded target molecules (Fig. 4b and Supplementary Movie 3).

These findings were further substantiated by the fact that antagonisation of the LiaRS response was only observed after addition of both, purified cell wall precursors and PG (Supplementary Fig. 6).

**DAP forms a tripartite complex with lipid II and PG**. To verify the formation of an antibiotic–target complex, purified lipid II was incubated with increasing concentrations of DAP in the absence and presence of PG, followed by extraction of the reaction mixture and subsequent thin layer chromatography (TLC) analysis. As expected, the formation of extraction-stable DAP complexes with lipid II was only observed in the presence of PG (Fig. 5a, left). Free PG, lipid II and DAP migrated to defined positions on the TLC. Increasing concentrations of DAP progressively diminished the amount of free, extractable lipid II as indicated by the respective band intensity compared to the control lacking DAP (Fig. 5a). In accordance with the formation of a tripartite complex, DAP and lipid II bands concurrently vanished from the TLC. DAP is known to aggregate and form oligomers of six or more molecules[14] and concordantly lipid II was almost

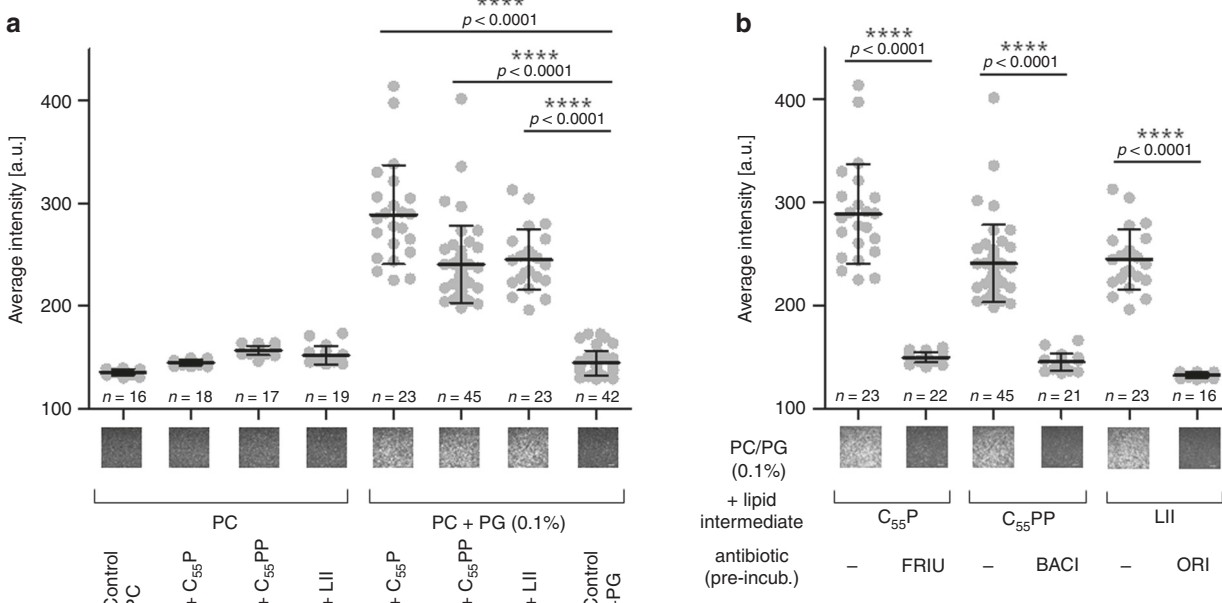

**Fig. 4 DAP binding to supported bilayers doped with cell wall lipid intermediates. a** Binding of DAP to supported bilayers is drastically increased in presence of bactoprenyl-coupled lipid intermediates and PG. Supported planar bilayers were prepared on coverslips using neutral DOPC lipids. Negatively charged PG lipids and bactoprenyl lipids were added, either 0.1 mol% bactoprenyl-coupled lipid intermediates or mixtures with PG (0.1 mol% each, 0.2 mol % PG served as a control). DAP was applied as a mixture of 1 µM native DAP and 50 nM DAP-FL. Movies with 100 frames were recorded at a frame rate of 60 Hz within 3 min after addition of DAP. The movies allowed to discern single binding events of DAP-FL. Exemplary fluorescence images are shown for each mixture. The field of view corresponds to $(32.5\,\mu m)^2$. **b** Inhibition of DAP-FL binding to supported planar bilayers (PC + 0.1% PG) by antibiotics that target specific bactoprenyl-coupled lipid intermediates. 300 nM of each antibiotic (FRIU, friulimicin; BACI, bacitracin and ORI, oritavancin) were incubated with the corresponding bactoprenyl-coupled lipid intermediate containing membranes for 5 min (antibiotic:lipid intermediates ratio 100:1). Excess of antibiotics was removed by buffer exchange followed by the addition of DAP as described in **a**. Exemplary movies are shown in Supporting Movies 2 and 3. Data were obtained from at least 20 movies for each experiment. We determined the mean values in a field of 160 µm² in the first image of each movie. Data in **a** and **b** are plotted as averages of these means and error bars represent the SD of all movies of a specific experiment. Significance was determined by unpaired Student's t-test with a 95% confidence interval, ****$p < 0.0001$. All experiments were independently repeated three times and yielded comparable results. Source data are provided as a Source Data file.

fully locked in an extraction-stable complex at a molar ratio of 1:10 with respect to DAP (Fig. 5a). In contrast, no change in band intensities was observed for lipid II and DAP in the absence of PG; even at the highest DAP concentration lipid II was uncomplexed (Fig. 5a, right). Complex formation was further not observed when lipid II and DAP were incubated in the presence of PC or the negatively charged cardiolipin (Supplementary Fig. 7), supporting specificity for PG to form a complex with DAP and lipid II.

Previous analyses showed only a minor impact of DAP on in vitro cell wall biosynthesis reactions, causing a 20% inhibition at a twofold molar excess of DAP[24,25]. However, these previous studies did not contain PG and did further not take into account that DAP oligomerises into multimers. In the presence of PG, the inhibitory effect of DAP on the MraY catalysed synthesis of lipid I was increased at a two-fold molar excess. MraY is the initial glycosyltransferase of peptidoglycan biosynthesis linking the first sugar building block to the lipid carrier $C_{55}P$ (Supplementary Fig. 3). The MraY catalysed reaction was almost completely blocked at a 10:1 molar ratio (Fig. 5b), in line with DAP oligomerisation. Similarly, the PBP2-catalysed transglycosylation of lipid II was inhibited in a dose-dependent manner (Fig. 5c).

## Discussion

DAP has served as a life-saving antibiotic for almost two decades. However, fundamental aspects of its killing activity, in particular molecular details, such as the identification of a specific molecular target, remained elusive[14]. In contrast, downstream effects triggered by DAP were extensively described and revealed unique features of its activity on the cellular level[9,12]. It became clear that DAP primarily impairs cell wall biosynthesis by targeting fluid microdomains, causing massive membrane rearrangements and displacement of cell wall biosynthetic enzymes. Such activities are reminiscent of cAMPs[12,26] and indeed DAP, in complex with $Ca^{2+}$, has been proposed to act like cAMPs[5]. Typically, most cAMPs act without specific molecular targets and may have minimal inhibitory concentrations in the millimolar range. The potency of DAP, in contrast, is much higher and compares well with the activity of those cAMPs that combine membrane effects with specific target binding, such as plectasin-like defensins or nisin-like lantibiotics, which may have MICs in the nanomolar range[27,28]. Both groups of cAMPs are known to bind to bactoprenyl-coupled cell wall precursors, specifically the peptidoglycan precursor lipid II, and it was tempting to assume similar activities for DAP, particularly since DAP elicits the LiaRS-mediated response specific for antibiotics interfering with the lipid II biosynthesis cycle[29,30]. However, unlike with nisin, plectasin, teixobactin and several other lipid II-binding antibiotics[18,25,28], we were previously unable to demonstrate such activities for DAP.

DAP is structurally related to the lipopeptides friulimicin and amphomycin, both of which are characterised by an overall negative charge. A DXDG or EF-hand motif is likely involved in $Ca^{2+}$ binding that is strictly required for antimicrobial activity of these antibiotics[4]. Despite such structural similarities, these true lipopeptides appear to significantly differ in their mechanism of action. Amphomycin and friulimicin bind to $C_{55}P$ and

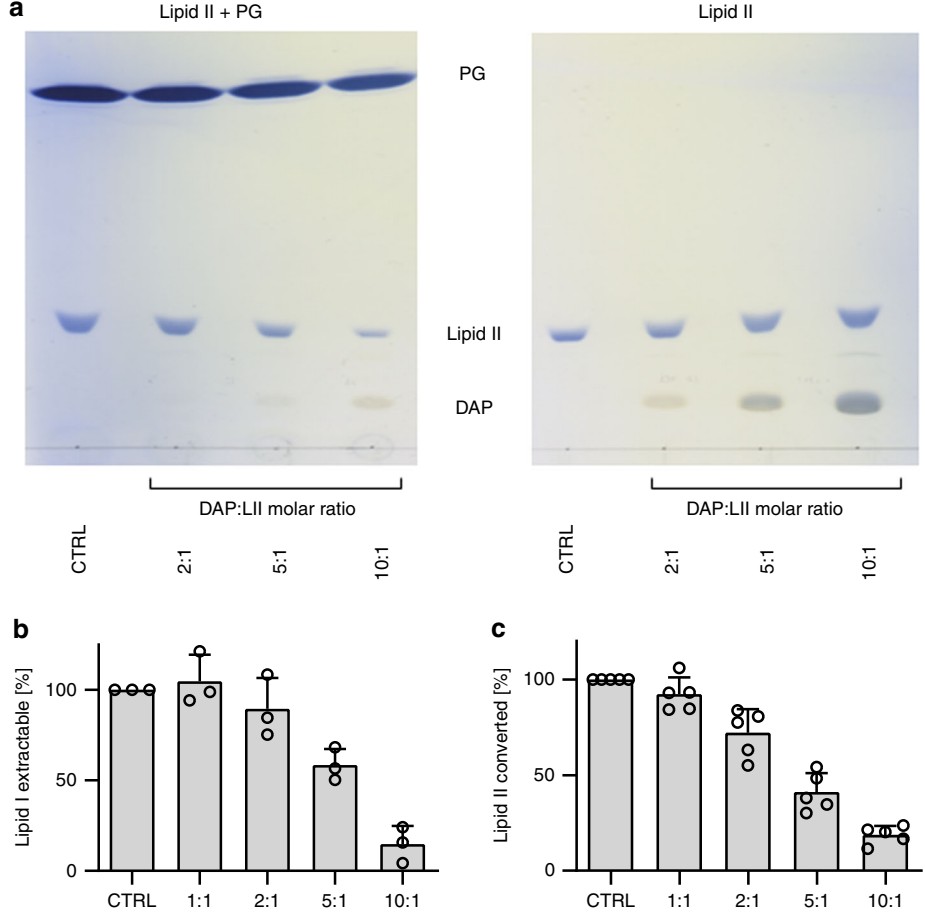

**Fig. 5 DAP forms a tripartite complex with lipid II and PG and inhibits cell wall biosynthesis in vitro. a** DAP was incubated with purified lipid II at increasing molar ratio in the presence (left TLC) and absence (right TLC) of PG. Reaction mixtures were extracted with BuOH and the upper solvent phase was spotted to TLC. Free PG, DAP and lipid II migrate to defined positions on the TLC, while components that are locked in complex are retained in the lower aqueous phase after extraction and as a result are diminished on the TLC. Representative images from five independent experiments are shown. **b** Impact of DAP on the MraY-catalysed synthesis of lipid I and **c** on the transglycosylation reaction catalysed by PBP2. DAP inhibits both reactions in a dose-dependent manner and almost completely blocks enzymatic activity when added in 10-fold molar excess. The enzymatic activity is expressed as synthesised lipid I (MraY) or converted lipid II (PBP2). The control reactions in the absence of antibiotics were set to 100%. DAP was added at molar ratios of 1:1 to 10:1 with respect to the lipid substrates as indicated. Data presented are means from three independent experiments and error bars represent the SD. Source data are provided as a Source Data file.

inhibit in vitro the formation of lipid I catalysed by the glycosyltransferase MraY at a two-fold molar ratio, corroborating the formation of active dimers[24,25]. Crystal structure analysis of another related calcium-dependent cyclic lipopeptide, tsushimycin, suggested that the biologically active form of the antibiotic is likely to be a dimer[31]. Dimerisation is dependent on the presence of $Ca^{2+}$, which results in a structure exhibiting a hydrophobic surface and a cleft-like tunnel, that is suitable for the accommodation of a target molecule such as $C_{55}P$.

Importantly, DAP has been reported to form oligomers[14] rather than dimers and was previously only tested at equimolar concentrations of $C_{55}P$ to DAP[25]. In addition, the detergent-based assays employed in the latter study lacked PG, which has been shown in numerous studies to be another prerequisite for DAP activity. We thus revisited bactoprenyl-coupled cell wall precursors as potential target molecules in a test system mimicking a more natural membrane environment and included higher DAP concentrations. Only in the presence of both $Ca^{2+}$ and PG an extraction-stable complex with lipid II was formed. Furthermore, binding of DAP to supported bilayers doped with bactoprenyl-bound precursors was strongly enhanced in the

presence of PG. Together these data reveal the formation of a tripartite $Ca^{2+}$-DAP complex with the anionic phospholipid and cell wall precursors.

It is well established that PG plays a key role in the mechanism of action of DAP, and PG levels have been associated with DAP susceptibility and resistance. It was recognised early that susceptible bacterial species, such as staphylococci are characterised by high membrane PG levels (up to 60%), while streptomycetes, including the DAP producer, have a reduced PG content[32]. DAP is active against Gram-positive bacteria and essentially lacks antimicrobial activity against Gram-negative species. Even in *E. coli* strains with a destroyed outer membrane barrier, no change in susceptibility was observed, which may well be linked to limited PG levels in these bacteria (about 15%)[33,34]. Numerous studies on clinical and laboratory daptomycin-resistant (DAP-R) strains, linked DAP non-susceptibility to the overall PG content[35–37]. Mutations in *mprF*, encoding for the bifunctional lysyl-phosphatidylglycerol (Lys-PG) synthase and flippase, are frequently found in DAP-R strains. Single nucleotide polymorphisms (SNPs) are often associated with increased conversion of PG to Lys-PG likely resulting in reduced DAP binding due to

charge repulsion[38]. A recent study hypothesised that mutations at the junction of synthase and flippase domain reduce intramolecular interactions proposed to result in an extended substrate spectrum of the flippase. Whether this directly affects translocation of the antibiotic or membrane-standing components crucial for activity remains to be elucidated[39].

Importantly, DAP exhibits limited toxicity in humans, correlating with the generally low abundance of PG in mammalian cells[40]. However, DAP was shown to be inactivated by pulmonary surfactant in vitro[41]. This effect is attributed to the relatively high PG content in lung surfactant (~10%), which excludes DAP as a therapeutic option in the treatment of pneumonia.

Our model membrane studies substantiate the proposed crucial role of PG for DAP membrane interaction, following structural transition and oligomerisation in the presence of $Ca^{2+}$[5,11,13]. Of note, an obviously defined stoichiometry in the binding of $Ca^{2+}$-DAP to PG of 1:1 or 1:2, depending on the model deployed, indicates a direct interaction, rather than PG-mediated modulating effects that may facilitate DAP membrane insertion indirectly[42,43]. In line, $Ca^{2+}$-DAP further induced the formation of extensive DAP/PG domains in giant unilamellar vesicles[44] and localised to membrane regions of increased fluidity (RIFs) in bacterial cells[12].

Anionic phospholipids, such as PG, are assumed to preferably localise to membrane domains in the septal and polar regions in bacterial cells, and to play important roles in biological processes. Phospholipids specifically interact with proteins and coordinate their spatial and temporal position and function, e.g. the positioning of the divisome complex at midcell, DNA replication, ATP synthesis and osmoregulation[45,46]. Furthermore, anionic phospholipids have been shown to be crucial for the enzymatic activity of membrane-bound and peripheral proteins, such as the initial glycosyltransferase MraY in peptidoglycan biosynthesis[47] and, thus, appear to impact on the organisation of cell wall biosynthetic machineries. In accordance, previous studies with Bacillus revealed that fluorescently labelled DAP preferentially localised at the division septum and in a helical pattern following the longitudinal axis of the cell, regions in the rod-shaped cells that are enriched in both anionic lipids, primarily PG, and $C_{55}P$-coupled cell wall precursors[9,29].

Moreover, association of phospholipids and bactoprenyl-bound cell wall precursors, like lipid II, has been reported[48]. Molecular dynamics simulations suggest that lipid II forms specific amphiphilic "island"-like regions on the membrane surface, in which the hydrophilic lipid II headgroup is central and surrounded by an extended and long-lived hydrophobic pattern. Importantly, the formation of this unique pattern induced by lipid II was observed only in membranes containing PG, but was absent in pure PC bilayers[48], creating an ideal "landing platform" for antibiotics.

DAP was shown to be bactericidal against both exponentially growing bacteria and S. aureus cells in stationary phase[49]. The authors at that time concluded that the bactericidal action of DAP likely does not require cell division or active metabolism. However, several lines of evidence showed that stationary-phase cells exhibit significant cell wall biosynthetic activity[50–52] and, thus, sufficient target availability. Reduced concentrations of cell wall precursors in stationary phase cells agree well with the increased concentrations (>100 µg ml$^{-1}$, 24 h) required to achieve comparable killing (three-fold log reduction) with regard to exponentially growing cells (2 µg ml$^{-1}$, 60 min). Moreover, due to its cationic nature, $Ca^{2+}$-DAP can bind to membranes even in the absence of molecular targets[42,43]. Likewise, the lantibiotic nisin also binds to membranes in the absence of lipid II[28]. This may also explain the fact that we found DAP-mediated killing is only delayed upon target blocking.

Membrane depolarisation has also been associated with the DAP killing mechanism, which is controversially discussed in literature[10,11,53]. Here, we show that initial binding of DAP to the septum correlates with antibiotic killing within the first 15 min of treatment (75% reduction in survival) and, thus, target binding appears the central event for promoting killing. In line with findings of Jung et al.[11] and the fact that pronounced depolarisation is observed only after 30 min of treatment[10], cell death is rather the consequence of lipid II binding, whereas depolarisation is most likely a result of disintegrative membrane rearrangements following PG and lipid II binding. The massive DAP clusters observed after prolonged incubation are highly reminiscent of the nisin:lipid II clusters reported by Hasper et al.[54].

Lipid II plays crucial roles in bacterial cell physiology that go beyond its role as a cell wall building block. In fact, lipid II appears to function as a crucial element in the organisation and coordination of enzymatic complexes, involved in peptidoglycan biosynthesis and cell division[21,22]. Thus, interfering with these membrane-bound machineries and biosynthetic networks by sequestration of lipid II can induce malfunctioning of a multitude of differential and overlapping cellular processes.

The identification of bactoprenyl-bound precursors as target molecules in this study sheds light on the unique requirements for DAP action. The interactions with these molecules and the cellular complexity arising from target binding can explain the pleiotropic cellular downstream effects of this unique antibiotic and reconcile the puzzling results from numerous studies on DAP resistance development (Fig. 6).

The identification of the molecular targets adds an important piece to the DAP MoA puzzle. However, since structural information (of comparatively simple complexes) on antibiotics in complex with full-length lipid II are scarce, the challenges of unravelling the structure and dynamics of the DAP tripartite complex will likely be much higher.

## Methods

**Chemicals.** All chemicals were of analytical grade or better. Bodipy FL succinimidyl ester, 6-(tetramethylrhodamine-5-(and-6)-carboxamido) hexanoic acid (TMR) succinimidyl ester, Bodipy FL Vancomycin and nile red were purchased from Thermo Fisher Scientific, Waltham, USA. DAP was labelled with Bodipy FL succinimidyl ester followed by ether precipitation, preparative TLC[55] and verification by MALDI-TOF spectrometry. The phospholipids 1,2-dioleoyl-sn-glycero-3-phosphocholine (DOPC), 1,2-dioleoyl-sn-glycero-3-phosphoglycerol (DOPG) and 1′,3′-bis[1,2-distearoyl-sn-glycero-3-phospho]-glycerol (DOCL, cardiolipin) were purchased from Avanti Polar Lipids (Alabaster, AL, USA) and used without further purification. Undecaprenyl phosphate ($C_{55}P$) and undecaprenyl diphosphate ($C_{55}PP$) were purchased from Larodan Fine Chemicals AB (Malmö, Sweden). DAP was purchased from Cubist Pharmaceuticals, Oritavancin was provided by The Medicines Company, Montreal, Canada and teixobactin was kindly provided by Kim Lewis, Northeastern University, Boston, USA. All other chemicals were from Sigma-Aldrich, Taufkirchen, Germany. Lipid II was enzymatically synthesised in vitro followed by n-butanol-pyridine acetate (pH 4.2) (2:1, v/v) extraction and purification via ion exchange chromatography[56].

**Strains used in this study.** S. aureus strains HG003[57] and RN4220[58] and Bacillus subtilis W168 sacA::pCHlux101 ($P_{liaI}$-lux)[17] were used. For cloning, E. coli DH5α (Invitrogen, Carlsbad, CA, USA) was used. To construct a strain expressing a FtsW-GFP fusion protein from the native FtsW promotor, the GFPmut2 coding sequence was amplified from plasmid pCQ11 (kindly provided by C. Quiblier and B. Berger-Bächi) using primers gfp-f (5′-AGTAACGCGTATGAGTAAAGGA GAAGAAC-3′) and gfp-r (5′-ATAGGCGCGAATTCTTTGTATAGTTCATCC-3′), the resulting fragment was restricted with EcoRI and MluI and ligated into pMAD[59], which has been cut with the same restriction enzymes giving plasmid pMAD-gfpmut2. A 506 bp fragment encoding the C-terminal part of ftsW and a 492 bp fragment downstream of ftsW were amplified using primers ftsW-up-f (5′-TACGCTAACAGATGGATCCCAATTCGAAT-3′), ftsW-up-r (5′-TGGCTAGTA TTTTTACGCGTAAATTGTCTTCT-3′) and ftsW-down-f (5′-GAAGACAATTT AATGAATTCAATACTAGCCAATA-3′) and ftsW-down-r (5′-TTTTATCTTAA AGATCTCTGTTTTGCTAT-3′), respectively, and the products cloned into pMAD-gfpmut2 using BamHI, MluI EcoRI and BglII. The final plasmid pMAD-ftsW-gfpmut2 was electroporated into S. aureus RN4220 and a described protocol[59]

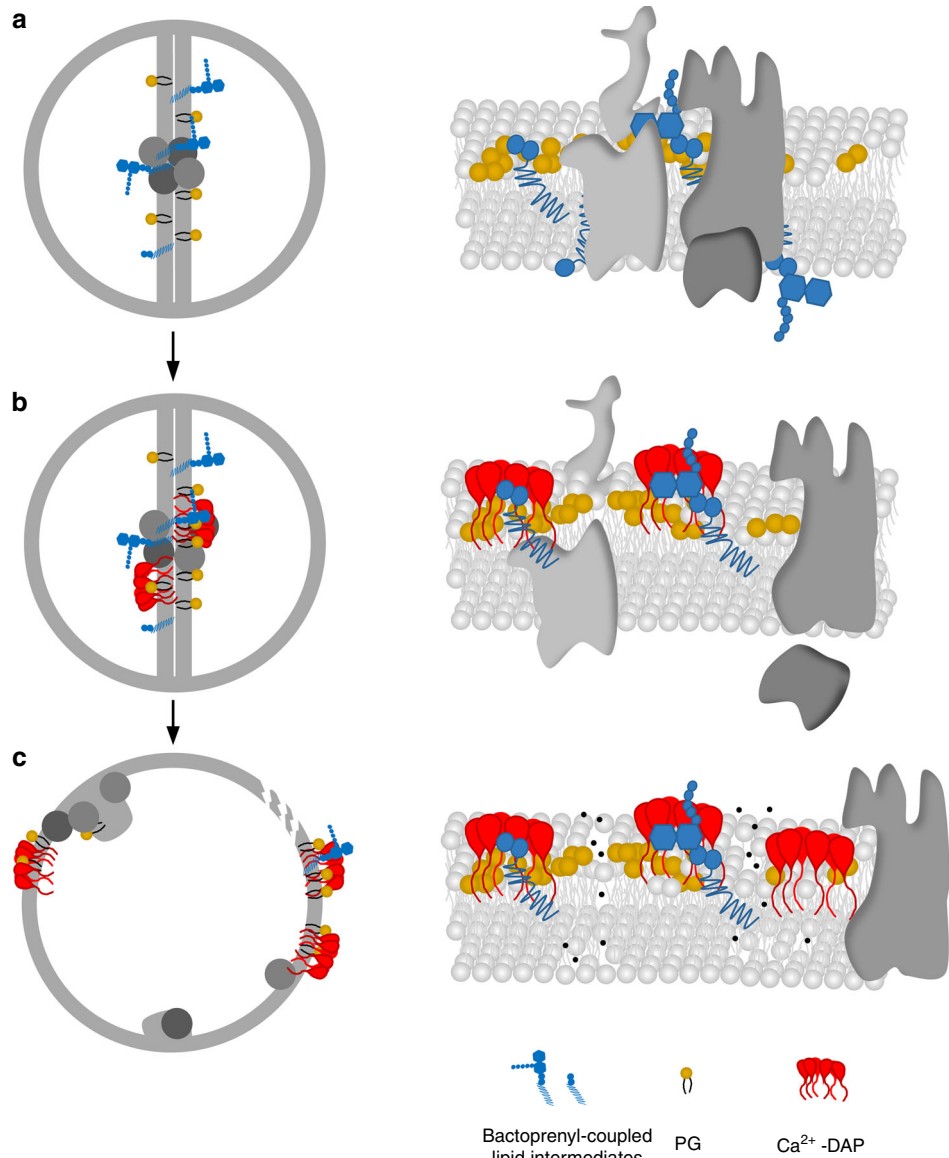

**Fig. 6 Proposed model for the mechanism of action of DAP. a** Orchestration of the cell wall biosynthesis machinery at the division septum of *S. aureus* in the absence of DAP. **b** $Ca^{2+}$-DAP oligomerises and preferentially localises at the division septum enriched in anionic lipids, primarily PG, and $C_{55}P$-coupled cell wall precursors. The formation of a tripartite complex of $Ca^{2+}$-DAP with PG and bactoprenyl-coupled lipid intermediates blocks cell wall synthesis and triggers the delocalisation of the cell wall biosynthetic machinery. **c** Prolonged treatment results in a progressive dispersion of DAP throughout the entire cytoplasmic membrane, followed by disintegration of the membrane bilayer finally resulting in membrane leakage and cell death.

was followed to achieve double homologous recombination of the plasmid insert into the genome.

**Luciferase reporter assays**. *B. subtilis* luciferase reporter assays were conducted as previously described[60]. Briefly, strains were grown in Mueller–Hinton broth at 30 °C containing 5 µg ml$^{-1}$ chloramphenicol until they reached an $OD_{600}$ of 0.5. Cells were added to 96-well white wall chimney plates containing antibiotics and luminescence measurements were performed at 30 °C in a microplate reader Spark 10M (Tecan). DAP and putative antagonists DOPG, $C_{55}P$ and lipid II were pre-incubated for 10 min prior to addition of the reporter strain. At least three independent biological replicate experiments were conducted. Data analysis was performed using Graph Pad Prism 5.01.

**Microscopy of bacteria**. To study the localisation of DAP, *S. aureus* HG003 was grown in LB to the mid-exponential phase ($OD_{600} = 0.5$) followed by the addition of 1.25 mM $CaCl_2$ and either unlabelled DAP or a mixture of labelled and unlabelled DAP (7 µg ml$^{-1}$ unlabelled/0.8 µg ml$^{-1}$ DAP-FL). To visualise cell membranes, nile red was added to a final concentration of 1 µg ml$^{-1}$. Cells were washed twice in LB, re-suspended and analysed by fluorescence microscopy. Therefore, cells were spotted onto microscope slides covered with a thin film of 1% agarose in

water. Microscopy was carried out at room temperature using a Zeiss Axio Observer Z1 microscope (Zeiss, Jena, Germany) equipped with HXP 120 V light source and an Axio Cam MR3 camera. Standard filter sets were used for Bodipy-FL (450–490 nm excitation and 500–500 nm emission), Marina blue (for native DAP visualisation, 335–383 nm excitation and 420–470 nm emission) and nile red (510–560 nm excitation and 590 nm long pass emission).

Image acquisition and analysis were performed with Zen 2 (Zeiss) and ImageJ v1.45s software (National Institutes of Health)[61]. For the quantification of fluorescence intensities, the mean grey values measured in the Bodipy-FL channel image of individual cells were summarised and divided by the total area of the cells measured in the phase contrast image. At least 350 cells were used for calculations for each condition.

**Killing kinetics**. To study the impact of short-term pre-incubation with teixobactin on *S. aureus* DAP susceptibility, cells were grown in LB to the mid-exponential phase ($OD_{600} = 0.5$) and the CFU ml$^{-1}$ of the culture was determined and set as 100%. Two aliquots were taken from the culture that were either left untreated or pre-incubated with teixobactin (four-fold MIC) for 2 min. Cells were spun down and re-suspended in LB supplemented with 1.25 mM $CaCl_2$ and DAP (10 µg ml$^{-1}$) followed by incubation at 37 °C. To study the effect of lipid II accumulation,

vancomycin (5 µg ml$^{-1}$) was added to cells grown to the mid-exponential phase which were then incubated for 30 min. Cells were washed twice in LB supplemented with 1.25 mM CaCl$_2$, DAP (5 µg ml$^{-1}$) was added and the cells further incubated at 37 °C. Controls were performed without pre-incubation with vancomycin and/or without the addition of DAP. Aliquots were taken at the indicated time points and CFU ml$^{-1}$ were determined. Therefore, cells were serially diluted in 0.9% NaCl solution and streaked on LB agar plates. Colony counts were determined after overnight incubation at 37 °C.

**Bilayer preparation**. In order to prepare planar bilayers on glass supports, very small unilamellar vesicles (VSUV) were prepared from detergent solution by addition of heptakis(2,6-di-O-methyl)-β-cyclodextrin (cyclodextrin) according to Roder et al.[62]. Neutral DOPC lipids were used to prepare fluid bilayers. Bactoprenyl-bound lipid intermediates and/or negatively charged DOPG were added with the amount of 0.1 or 0.2 mol%, respectively. For each mixture, 1.3 mM lipid solution was prepared in chloroform, and the chloroform was slowly removed in a nitrogen stream. The resulting lipid film was solubilized in HEPES buffer (20 mM HEPES, 150 mM NaCl, pH 7,4) supplemented with 20 mM Triton X-100. The lipid-detergent stock solution of 5 mM lipid was stored at −4 °C until use. A second stock solution contained 4 mM cyclodextrin in HEPES buffer and was also stored at −4 °C. A suspension of VSUV was prepared by first diluting the lipid-detergent stock solution 10-fold in HEPES buffer, followed by addition of an equal volume of cyclodextrin stock solution and immediate, thorough mixing by vortexing for 2 min. Vesicles were generally used within 1 h after preparation.

Coverslips (18 × 18 mm) were cleaned overnight in fresh Piranha solution (one part H$_2$O$_2$ 30% and two parts concentrated H$_2$SO$_4$), rinsed thoroughly with milliQ water and dried in a nitrogen stream. Clean coverslips were placed into custom-built sample chambers with a Teflon O-ring as seal and two metal clips to fix the metal insert on top of the coverslip.

Bilayers were prepared immediately by adding 400 µl freshly made vesicle suspension filling the well of the sample chamber. Due to electrostatic interaction between the lipid headgroups and the highly hydrophilic glass surface, vesicles readily attached to the coverslip. The high surface tension led to fusion of VSUV and formation of homogeneous bilayers on the complete cover slip within 5 min. Residual, non-fused vesicles were removed by carefully washing three times with HEPES buffer. During the washing steps care was taken to not dry out the lipid bilayer. After washing, bilayers were kept in HEPES buffer.

**Bilayer binding assay and dual colour HILO imaging**. Successively, 1.25 mM CaCl$_2$ and DAP—in a mixture of 1 µM native DAP and 50 nM Bodipy FL-labelled daptomycin (DAP-FL)—in HEPES buffer were added to freshly prepared bilayers. For the binding inhibition test, 300 nM of the antibiotics friulimicin, bacitracin and oritavancin (specifically targeting C$_{55}$P, C$_{55}$PP and lipid II, respectively) were first incubated with the corresponding bactoprenyl lipid containing membranes for 5 min. Bacitracin was added together with 0.5 mM ZnCl$_2$ and oritavancin together with 0.002% Tween 80 (v/v). The peptide to lipid ratio was 100:1. Excess of antibiotics was removed by buffer exchange before CaCl$_2$ and DAP addition.

Images were acquired at a custom-built, single molecule sensitive, inverted microscope capable of TIRF microscopy with an EMCCD camera (iXon DU-897D; Andor Technology, Belfast, Northern Ireland)[63] using Andor Solis software. Lasers and acousto-optic tunable filter (AOTF, Gooch&Housego) were controlled by Labview 2012 software. Illumination with total internal reflection reduced fluorescence excitation to a thin region at the coverslip surface with the benefit of background suppression from fluorescence outside the illuminated region. The illumination beam angle was adjusted by tilting a collimated laser beam in the object focal plane of the imaging lens, which focused the beams in the back focal plane of the objective, until total reflection at the coverslip/medium interface was reached. Using a ×63 objective lens with a NA 1.45 (Zeiss), additional ×4 magnification and 2 × 2 binning during acquisition resulted in a pixel size of 127 nm. The imaged field of view with the extension (32.5 µm)$^2$ corresponded approximately to 1/4 of the total illuminated field. Thus, it can well be assumed that the observed field showed an isotropic fluorescence, and that there was no influx of putatively unbleached and no efflux of putatively bleached molecules. This assumption is corroborated by an evaluation of the movie data, which did not show any spatial fluorescence gradient occurring over the field of view as a function of time. DAP-FL was excited with laser light at 488 nm (Sapphire-100; Coherent, Santa Clara, CA) and fluorescence was detected by use of a standard fluorescein dichroic and a notch filter (NF03-488E-25, Semrock). Movies with 100 frames were recorded at a frame rate of 60 Hz immediately after addition of DAP within 5 min.

In each data acquisition session, all samples were measured during one day in order to guarantee optimal comparability among the samples. From each sample at least 20 movies at different locations of the bilayer on each cover slip were acquired. Intensity values over a cropped region of 100 × 100 pixels corresponding to 160 µm$^2$ within the first image of each movie were averaged in order to circumvent effects of possible photobleaching or molecular transport during movie data acquisition. Data analysis was performed using Origin 2019 software. The average background signal of the camera was subtracted and the intensity values were averaged for each experimental condition separately. The observation field was defined by the camera position and was constant, however, small deviations in the very sensitive TIRF excitation beam path resulted in absolute intensity variations from day to day. A Student's t-test was performed to demonstrate the significance of the intensity differences between respective samples from one session. All experiments were repeated three times and yielded comparable results.

The same instrument was employed for fluorescence microscopy based on HILO illumination[64] by reducing the incidence angle of the laser beam in the back focal plane of the objective lens compared to that for TIRF excitation[65]. Illumination using HILO reduced fluorescence excitation to a sheet-like axial section near the coverslip surface resulting in suppression of fluorescence outside the illuminated section. For dual-colour imaging of bacteria, an image splitter (OptoSplit II, Cairn Research, Faversham, UK) was inserted in the detection pathway in front of the camera. A dichroic mirror (FF509-FDi01-25 × 36, Semrock, Rochester, USA) and a band-pass emission filter (LP02-561RE-25, Semrock) were used to separate the emission channels. A motorised stage controlled by µManager[66] was used to acquire z-stacks with a step size of 50 nm. The stacks were then filtered and reconstructed in three-dimensions by the 3D Hybrid Median Filter function of ImageJ.

**Complex formation of DAP with purified lipid II**. Complex formation was analysed by incubating 3 nmol of purified lipid II in 50 mM Tris–HCl pH 7.5 and 1.25 mM CaCl$_2$ in presence or absence of 75 µg DOPG with increasing concentrations of DAP for 20 min at room temperature. Complex formation was further tested with 75 µg DOPC and DOCL. Free lipid II molecules were extracted with n-butanol–pyridine acetate (pH 4.2) (2:1, v/v) and analysed by TLC using chloroform/methanol/water/ammonia (88:48:10:1, v/v/v/v) as the solvent and phosphomolybdic acid (PMA) staining. The quantitative analysis of lipid II extracted to the butanol phase was carried out using ImageJ. Experiments were performed in triplicate.

**Impact of DAP on peptidoglycan biosynthesis reactions in vitro**. Peptidoglycan synthesis reactions were reconstituted in vitro using purified proteins and substrates. PBP2-His$_6$ and MraY-His$_6$ were produced in E. coli and purified via IMAC after solubilisation with 0.06% Triton and 18 mM n-dodecyl-β-D-maltoside, respectively[25]. Enzymatic activity of purified MraY-His$_6$ was assessed in a total volume of 50 µl containing 5 nmol C$_{55}$P and crude substrate in 0.08% Triton X-100, 37.5 µg DOPG, 100 mM Tris–HCl, 30 mM MgCl$_2$, pH 8.0, and 1.25 mM CaCl$_2$. The reaction was initiated by the addition of the recombinant enzyme and extracted after incubation for 90 min at 30 °C.

Transglycosylation by PBP2 was determined by incubating 3 nmol of lipid II in 0.04% Triton X-100, 20 mM MES, 2 mM MgCl$_2$, 2 mM CaCl$_2$ pH 5.5 in a total volume of 50 µl. The reaction was initiated by the addition of 8 µg of PBP2-His$_6$ and incubated for 2 h at 30 °C.

DAP was added in molar ratios ranging from 1 to 10 with respect to the respective substrate (C$_{55}$P or lipid II) in all in vitro assays and pre-incubated for 10 min prior to addition of the enzyme. Lipid intermediates were extracted from the reaction mixtures with n-butanol/pyridine acetate, pH 4.2 (2:1, v/v) and analysed by TLC as described above. Quantification was carried out by phosphorimaging in a StormTM imaging system (GE Healthcare) or PMA staining and analysis performed using ImageJ and Graph Pad Prism. Experiments were performed at least in triplicate.

**Accumulation of lipid II**. To verify accumulation of lipid II after vancomycin pre-treatment, cells were spun down and resuspended in phosphate buffered saline (PBS). Lipid extraction and TLC analysis were performed as described for the in vitro assays with the exception that the BuOH phase was washed twice with acidified H$_2$O.

**Reporting summary**. Further information on research design is available in the Nature Research Reporting Summary linked to this article.

## Data availability
The authors declare that all data supporting the findings of this study are available within the Article and its Supplementary Information. The source data underlying Figs. 1a, 2b, 3b, c, 4a, b, 5a–c and Supplementary Figs. 1, 5a, c, d, 6a, b and 7 are provided as Source Data file.

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

## Acknowledgements

Funding was provided by the Deutsche Forschungsgemeinschaft (DFG, German Research Foundation)—Project-ID 398967434—TRR 261 (to T.S., F.G., U.K.) and grant SCHN1284/2 (to T.S.), NIH (grant 5R01AI13262701 (to T.S. and F.G.)), and the German Center for Infection Research (DZIF) (to F.G.). K.M.S. was supported by a fellowship from the Cusanuswerk. We would like to thank Kim Lewis for providing teixobactin and Julia Deisinger for PBP2-His$_6$, Michaele Josten for mass spectrometry, and Thorsten Mascher for providing strain TMB1617 *B. subtilis* W168 s*acA*::pCHlux101 (P$_{liaI}$-*lux*). We thank Kenneth Pfarr for proofreading.

## Author contributions

U.K. and T.S. designed and coordinated the overall study. The experiments were performed by F.G., A.M., K.M.S., X.L., K.C.L., A.K. and M.S.; F.G., A.M., K.M.S., U.K. and T.S. analysed the data. The manuscript was written by F.G., A.M., H.-G.S., U.K. and T.S. with input from all authors.

## Competing interests

The authors declare no competing interests.
