## [Peer Review File · Nature Communications]

Reviewers' comments:

Reviewer #1 (Remarks to the Author):

The study of Grein et al represents a major scientific breakthrough – finally the mode of action of daptomycin, one of the most important antibiotics in clinical use, is revealed. The authors demonstrate that daptomycin forms a tripartite complex including the anionic phospholipid phosphatidylglycerole (PG) to inhibit cell wall biosynthesis by blocking the central cell wall precursor carrier lipid II. The study is sound, based on a broad set of elegant experiments, and well written. I have only some minor comments:

1. Results, beginning of first paragraph: I does'n get clear which statements refer to previous findings and at which point the description of results from the current study start. Please reword.
2. Lines 126 and following: Please indicate if teixobactin was used at sub-bactericidal concentrations. Otherwise, how could teixobactin- and daptomycin-mediated killing be distinguished?
3. Line 153-154: Was lipid II content increased here by vancomycin pretreatment as described above? Please clarify.
4. Page 7, first paragraph: Did the authors test other bacterial phospholipids in addition to PG for their capacity to form complexes with daptomycin and lipid II? Otherwise, they should tone down their statements on the specific influence of PG on daptomycin activity in the discussion.
5. Line 193: Please explain MraY
6. Lines 218.220: Are all mentioned lipopeptide antibiotics, friulimicin and amphomycin, Ca-dependent as for daptomycin?
7. Discussion: This study sheds new light on the recent finding that spontaneous daptomycin resistance involves mutations in the phospholipid flippase MprF in *S. aureus*. Is it possible that mutated MprF might translocate daptomycin-bound phospholipids to cause resistance?
8. Fig. 3 b: Please indicate to which bar pairs the indicated p values refer. Other bar graphs: please check for which of them it would be suitable to indicate significances.

Reviewer #2 (Remarks to the Author):

The manuscript by Grein et al. attempts to identify the molecular target of daptomycin in the cell membrane. The authors propose the formation of a "tripartite complex" between daptomycin, phosphatidylglycerol (PG), and undecaprenyl (UDP)-coupled cell envelope precursors. The authors primarily use in vitro membrane reconstitution and fluorescence microscopy approaches to evaluate their claim. As has been noted previously, the authors show that daptomycin activates the cell membrane stress response and binds at the division septum. The authors present data to show that less daptomycin is bound to the division septum when *S. aureus* is pre-incubated with antibiotics that associate with the lipid portion of UDP-intermediates. Using in vitro bilayers supplemented with PG and bactoprenyl lipids, they show less daptomycin binding when pre-incubated with antibiotics that target bactoprenyl-containing lipids. The authors use thin layer chromatography to examine potential interactions between these three compounds. Although the authors' claims that daptomycin, phosphatidylglycerol, and undecaprenyl-coupled intermediates form a tripartite complex as part of the mechanism of action of the antibiotic is certainly an attractive hypothesis, the data presented appear to be preliminary to substantiate this claim, for the following reasons:

1. In large part, the data in Figures 1, 2, and 4, recapitulate what is known regarding the binding and effects of daptomycin, namely that it activates the LiaRS stress response (PMID: 19164152, PMID: 21986832, PMID: 26495887), binds at the division septum (PMID: 22661688, PMID:

23882013), and leads to a time dependent loss of membrane integrity and cell death (PMID: 1687346, PMID: 27791134). The authors spend a significant proportion of this manuscript showing experiments that only confirm other previous observation. These data could be referenced and the figures added as supplementary material.

2. Figure 6a demonstrates phosphatidyl glycerol is important for daptomycin binding, but the conclusions regarding lipid II are less clear. First, the important controls of lipid II plus PG, without added daptomycin, are not shown. This is critical to exclude the possibility of a PG-lipid II complex forming and removing lipid II in the absence of daptomycin. Second, the control of PG plus daptomycin is not shown, to establish the degree of binding in the absence of lipid II in this assay. I assume calcium was included in this assay, though the concentration was not specified.

3. Further, in Figure 6 panels b) and c), enzymatic activity of MraY and PBP2 are shown to be inhibited by daptomycin at high molar ratios. Previously, work using this technique (PMID: 19164139) showed friulimicin B bound C55-P in an approximately equimolar (1:1) ratio, and did not inhibit downstream reactions (MurG, FemX, PBP2), suggesting specificity for C55-P of this compound. In this same study daptomycin at equimolar concentrations did not inhibit conversion of C55-P to lipid II. In the current work it is not until daptomycin:lipid molar ratios exceed 5-10:1 that we see an effect on lipid I synthesis and lipid II utilization. If daptomycin was forming a complex with undecaprenyl-intermediates, it is unclear why such a high molar ratio is needed to observe any effects? This finding seems to argue against a specific interaction, and rather describes a more general consequence of effects on the membrane. This would be consistent with what this group has previously shown, namely that daptomycin sequesters fluid lipids (including bactoprenol?) and displaces membrane associated proteins (PMID: 27791134), leading to inhibition of cell wall synthesis.

4. From the data presented in Figure 5, the authors suggest that the addition of oritavancin, bacitracin, and friulimicin block daptomycin binding to target membranes by blocking association with lipid intermediates. Did the authors exclude changes in membrane physiochemical properties, such as shifts in fluidity, as a potential source of decreased binding?

5. Figure 3 demonstrates decreased daptomycin binding after pre-incubation with the lipid II-binding antibiotics teixobactin and oritavancin. I am unsure why they authors did not include the vancomycin data here as well. I would suggest incorporating the vancomycin data from Supplementary Figure 5 into the main Figure 3, using the same conditions and time points in both experiments.

6. Further detail involving the methodology of Figure 5 would be helpful. Is the same location of fluorescence (over time, between samples) used? What percentage of the field of view over total area fluorescing? Were any efforts taken to determine the rate of diffusion of substances into the field of fluorescence over time? Lipid I might be useful to include as well. Further description of statistical analysis performed should be included in methods and appropriate statistical values should be marked in the Figure.

7. Supplementary Figure 6: Please specify the molar concentrations used in the experiment. If a 1:1 ratio of DAP:DOPG was used, and 1:1:1 of DAP:DOPG:C55-P, then the second condition would be 1 part DAP to 2 parts lipid (DOPG, C55-P), and may result in the observed decrease in fluorescence. Was a 1:0.5:0.5 mixture used?

Minor Comments

1. The introduction has many overstated statements on the use of daptomycin and the clinical impact of infections treated with this antibiotics. Daptomycin is really not a last resort drug for staphylococci.

2. The manuscript would benefit from revision of grammar as there are errors regarding comma

usage, tense, sentence fragments and run-on sentences.

3. The text frequently refers to daptomycin potentially binding to Lipid II (major comment above), however Figure 5 demonstrates potential strongest binding effect to C55P and then C55PP. Although the manuscript title reflects this, would alter wording throughout the manuscript to reflect this.

4. In Figure 1, the concentrations of daptomycin differ from what is written in the corresponding section in the Materials and Methods.

5. Supplementary Figure 3 is superfluous, a citation directing the reader to its source (PMID: 23215859) would be sufficient.

6. Would include fluorescence intensity in Figure 3b.

7. In Supplementary Figure 4, I believe the middle column of images should be labeled "FtsW-GFP" not "DAP-FL."

Response to referees

Reviewer #1 (Remarks to the Author):

The study of Grein et al represents a major scientific breakthrough – finally the mode of action of daptomycin, one of the most important antibiotics in clinical use, is revealed. The authors demonstrate that daptomycin forms a tripartite complex including the anionic phospholipid phosphatidylglycerole (PG) to inhibit cell wall biosynthesis by blocking the central cell wall precursor carrier lipid II. The study is sound, based on a broad set of elegant experiments, and well written. I have only some minor comments:

We thank this reviewer for the careful review, assessment and helpful comments.

1. Results, beginning of first paragraph: I does'n get clear which statements refer to previous findings and at which point the description of results from the current study start. Please reword.

The section has been reworded to clearly differentiate between previous findings and results from the current study. The paragraph (page 4, line 101) now reads: "DAP had previously been demonstrated to elicit the LiaRS stress response, supporting a specific inhibition of cell wall biosynthesis. The LiaRS two component system (TCS) is known to respond to antibiotics that interfere with the lipid II biosynthesis cycle^{15,16}. In search for a molecular target within the peptidoglycan biosynthesis pathway we here revisited the effect of DAP on the LiaRS response, but monitored *lial-lux* induction over time and tested the impact of Ca²⁺." The reference referring to the *lia* bioreporter strain has been shifted to the following more general sentence to make it even clearer.

2. Lines 126 and following: Please indicate if teixobactin was used at sub-bactericidal concentrations. Otherwise, how could teixobactin- and daptomycin-mediated killing be distinguished?

Cells at OD₆₀₀=0.5 were pre-treated with teixobactin (4-fold MIC) for only 2 min, subsequently washed and no killing was observed. To highlight that under these conditions no killing is observed by TEIX only treatment we now included a control of TEIX pre-treated cells (black curve) and added a colored legend to the Figure 3c. A corresponding sentence was added to the figure legend: "Bactericidal effects were not observed in cells only pre-treated with TEIX (black line).".

Please also note that we show % survival instead of % killing. Importantly, only in teixobactin pre-treated cells a "protective effect" towards DAP mediated killing is observed (blue curve; delaying DAP induced killing (red curve)).

3. Line 153-154: Was lipid II content increased here by vancomycin pretreatment as described above? Please clarify.

Lipid II content was increased by VAN pre-treatment (treatment for 30 min, sub-lethal concentration) analogous to Suppl. Figure 5a. We rephrased this sentence to make clearer that increased lipid II (target) levels induced by VAN pre-treatment enhance DAP mediated killing. The sentence (page 6, line 155) now reads: "Pre-incubation with vancomycin for 30 min resulted in strongly elevated cellular lipid II levels triggering hyperaccumulation of DAP (Supplementary Figure 5a-c)..". We are sorry for the word "washing" that mistakenly remained in the sentence and apparently caused confusion. This has been corrected.

4. Page 7, first paragraph: Did the authors test other bacterial phospholipids in addition to PG for their capacity to form complexes with daptomycin and lipid II? Otherwise, they should tone down their statements on the specific influence of PG on daptomycin activity in the discussion.

In addition to PG, the impact of the neutral phosphatidylcholine (PC) was tested using supported bilayers and shown not to affect DAP binding (new Fig. 4, previous Fig. 5). To provide a more comprehensive picture we now include data on the capacity of additional phospholipids to form complexes with lipid II and DAP. As the new Supplementary Figure 7 shows, complex formation of DAP with lipid II is observed with PG, but not with PC or the negatively charged cardiolipin (CL). These data are now mentioned in the main text (page 8, line 198 ff): “Complex formation was further not observed with PC or the negatively charged cardiolipin (CL) (Supplementary Figure 7), supporting specificity for PG to form a complex with DAP and lipid II.” The method section has been changed accordingly.

5. Line 193: Please explain *MraY*

We have rephrased the sentence and included information that *MraY* is the initial glycosyltransferase of peptidoglycan biosynthesis (page 8, line 206): “*MraY* is the initial glycosyltransferase of peptidoglycan biosynthesis linking the first sugar building block to the lipid carrier C₅₅P (Suppl. Fig. 3).”

6. Lines 218.220: Are all mentioned lipopeptide antibiotics, friulimicin and amphomycin, Ca-dependent as for daptomycin?

The activity of all lipopeptides mentioned depends on the presence of Ca²⁺. We highlighted this in the text (page 9, line 232): “DAP is structurally related to the lipopeptides friulimicin and amphomycin, both of which are characterized by an overall negative charge. A DXDG or EF-hand motif likely involved in Ca²⁺ binding is strictly required for antimicrobial activity of these antibiotics³.”

7. Discussion: This study sheds new light on the recent finding that spontaneous daptomycin resistance involves mutations in the phospholipid flippase *MprF* in *S. aureus*. Is it possible that mutated *MprF* might translocate daptomycin-bound phospholipids to cause resistance?

This is a very interesting hypothesis, and this possibility was more recently highlighted (PMID: 30563904). We now refer to *MprF* in the discussion (page 10, lines 264ff) and cite the corresponding literature (Ernst et al., MBio, 2018 and Yang et al., AAC, 2013). The paragraph now reads: “Mutations in *mprF*, encoding for the bifunctional lysyl-phosphatidylglycerol (Lys-PG) synthase and flippase, are frequently found in DAP-R strains. SNPs are often associated with increased conversion of PG to Lys-PG likely resulting in reduced DAP binding due to charge repulsion (Yang et al., 2013). A recent study hypothesized that mutations at the junction of synthase and flippase domain reduce intramolecular interactions proposed to result in an extended substrate spectrum of the flippase. Whether this directly affects translocation of the antibiotic or membrane-standing components crucial for activity remains to be elucidated (Ernst et al., 2018). “

8. Fig. 3 b: Please indicate to which bar pairs the indicated p values refer. Other bar graphs: please check for which of them it would be suitable to indicate significances. Figure 3b has been changed to make clear to which bar pairs the p values refer. Data presentation

was further optimized: In contrast of showing percentage of control values we now show mean gray values per cell and thus show the control. Box plots represent the interquartile range of the data and the whiskers represent minimal and maximal values in the new figure.

To indicate significance p values have further been added to Figure 5.

Reviewer#2 (Remarks to the Author):

The manuscript by Grein et al. attempts to identify the molecular target of daptomycin in the cell membrane. The authors propose the formation of a “tripartite complex” between daptomycin, phosphatidylglycerol (PG), and undecaprenyl (UDP)-coupled cell envelope precursors. The authors primarily use in vitro membrane reconstitution and fluorescence microscopy approaches to evaluate their claim. As has been noted previously, the authors show that daptomycin activates the cell membrane stress response and binds at the division septum. The authors present data to show that less daptomycin is bound to the division septum when *S. aureus* is pre-incubated with antibiotics that associate with the lipid portion of UDP-intermediates. Using in vitro bilayers supplemented with PG and bactoprenyl lipids, they show less daptomycin binding when pre-incubated with antibiotics that target bactoprenyl-containing lipids. The authors use thin layer chromatography to examine potential interactions between these three compounds. Although the authors’ claims that daptomycin, phosphatidylglycerol, and undecaprenyl-coupled intermediates form a tripartite complex as part of the mechanism of action of the antibiotic is certainly an attractive hypothesis, the data presented appear to be preliminary to substantiate this claim, for the following reasons:

1. In large part, the data in Figures 1, 2, and 4, recapitulate what is known regarding the binding and effects of daptomycin, namely that it activates the LiaRS stress response (PMID: 19164152, PMID: 21986832, PMID: 26495887), binds at the division septum (PMID: 22661688, PMID: 23882013), and leads to a time dependent loss of membrane integrity and cell death (PMID: 1687346, PMID: 27791134). The authors spend a significant proportion of this manuscript showing experiments that only confirm other previous observation. These data could be referenced and the figures added as supplementary material.

We apologize that, at first glance, the mentioned data presented in our work do not emphasize sufficiently on the novel findings obtained. Although applying approaches previously used, this study includes novel details which are of utmost importance for the quintessence of our work and finally the identification of the molecular target.

This said, it is certainly not our aim to recapitulate what others have done and it is also not our intention to hide what has been done before (as demonstrated by the literature cited and the discussion). Rather it has been our aim to consolidate our findings and previous knowledge. Working on the mechanism of action of this antibiotic (and related lipopeptides) ourselves for more than 15 years we found that a critical aspect is that each lab uses different model organisms, different techniques and last but not least different antibiotic and varying Ca^{2+} concentrations, which resulted in a variety of explanations for the mode of action of daptomycin. In addition, the variety of experiments, experimental conditions and model organisms used in the different studies strongly hampered a direct and unrestricted data comparison. To this end we decided to revise key experiments (reflecting key findings; shown in Figures 1,2 and 4) and combine those with novel experimental set ups/conditions relevant to the current paper.

Of note, our experiments further add significant value to previous knowledge as outlined below in more detail:

- **Specific comment to “liaRS response”:** In the publication mentioned by this reviewer (PMID:19164152) microarray analyses showed induction of the liaRS stress response. However, the data presented in this previous study represent endpoint measurements, and do furthermore not address the effect of Ca²⁺. In contrast, we monitored the DAP mediated effect on LiaRS over time in the absence and presence of Ca²⁺. The effects observed in our study already point towards a more specific effect on the lipid II biosynthesis cycle, compared to what was previously shown.
Regarding the other two references cited by this reviewer: PMID 21986832 doesn't present data on LiaRS and PMID 26495887 is a review article citing PMID 19164152.
- **Specific comment to “binding to the division septum”:** Both studies mentioned by this reviewer use either *Bacillus* (PMID 22661688) or *Enterococcus* (PMID 23882013) as a model organism. We here used *S. aureus* as a model and more importantly we show here that native, unlabeled DAP localizes to the division septum, revealing that this is not a Bodipy-FL mediated effect. We further show co-localization with a putative lipid II flippase FtsW in addition (the localization of FtsW to the division septum has only recently (August 2019) been published in a Nature Microbiology paper by Mariana Pinho).
- **Comment on “time dependent loss of membrane integrity and cell death”:** Our data do not focus on time dependent loss of membrane integrity and cell death. Instead we focus on the biphasic binding of DAP over time which has, to the best of our knowledge, not been subject of any other study before. We show for the first time that the initial septal binding of DAP correlates with killing. This has been extensively discussed in the context of previous work. The references mentioned by this reviewer, in our opinion, do not fit into the context, as we are not focusing on the impact on membrane integrity. PMID 1687346: This is one of the earliest papers investigating the effect of DAP on membrane potential and amino acid transport. PMID 27791134: This is our own work published in 2016 in PNAS. This work was performed using *B. subtilis* and we could demonstrate gradual depolarization after prolonged incubation with DAP. However, in our current study we show that the initial septal binding is most relevant for killing. Of note, contrary to the different membrane-perturbing mechanisms proposed for DAP, we could not detect leakage of specific cations (PMID 27791134). Importantly, in this previous study we found that only higher (lytic) DAP concentrations resulted in a gradual membrane depolarization after 25 minutes revealing that DAP does not form distinct membrane pores. These results are perfectly in line with the data presented in the current study revealing that loss of membrane integrity and cell death are downstream effects resulting from blocking undecaprenyl-coupled lipid intermediates. We discuss this on page 12, lines 315 ff.

To avoid the impression that “this manuscript showing experiments that only confirm other previous observation” and concurrently highlight novel aspects relevant to the study, we revised the results section (e.g. page 4, lines 91 ff) and link our findings to previous knowledge at several occasions.

In addition, and in light of the comments of this reviewer in general (as specified here and below), we strongly contradict the evaluation of our work as “preliminary”.

2. Figure 6a demonstrates phosphatidyl glycerol is important for daptomycin binding, but the conclusions regarding lipid II are less clear. First, the important controls of lipid II plus PG, without added daptomycin, are not shown. This is critical to exclude the possibility of a PG-lipid II complex

forming and removing lipid II in the absence of daptomycin. Second, the control of PG plus daptomycin is not shown, to establish the degree of binding in the absence of lipid II in this assay. I assume calcium was included in this assay, though the concentration was not specified.

We agree that this control is most critical, and thus it was of course already included in the submitted version of our manuscript (previous Figure 6a, lane 1; named control 0:1). As shown, lipid II is not removed in absence of DAP. This is only observed at increasing concentrations (in the presence of PG). In order to highlight the control we optimized labeling of the new Figure 5a.

Regarding the second control requested: It is well known that PG and DAP interact as revealed in dozens of papers by us and others, which have been referenced and discussed. This interaction is one aspect relevant for DAP resistance mechanisms. In addition, this is also the reason for the inactivation of DAP by lung surfactant, which contains substantial amounts of PG (these facts are mentioned in our manuscript and cited accordingly).

However, and that is the important point here, the interaction with PG alone does not explain the specificity for bacterial cells. As our data show, this specificity is mediated by the interaction with undecaprenyl phosphate-coupled lipid intermediates such as lipid II. Thus, the focus of this experiment was on the interaction of DAP with lipid II (in presence and absence of PG). That the interaction of DAP with PG is most relevant in the presence of undecaprenyl phosphate-coupled lipid intermediates and particularly only when both are incorporated in bilayers is shown in the new Figure 4 (previous Figure 5): Here, the PG control reveals that PG alone does only marginally affect DAP binding, compared to binding to bilayers doped with both PG and lipid intermediates. In addition, these results are further supported by the antagonization of the LiaRS response (Supplementary Figure 6).

Following the recommendation of reviewer 1, we now further include data showing that complex formation of DAP with lipid II is observed with PG, but not with PC or the negatively charged cardiolipin (CL). These data are now included in a new Suppl. Figure 7 and the main text (page 7, line 184 ff): "Complex formation was further not observed with PC or the negatively charged cardiolipin (CL) (Suppl. Figure 7), supporting specificity for PG to form a complex with DAP and lipid II."

Ca²⁺ was added at a concentration of 1.25 mM, corresponding to the concentration in serum. This information has been added to the material and method section.

3. Further, in Figure 6 panels b) and c), enzymatic activity of MraY and PBP2 are shown to be inhibited by daptomycin at high molar ratios. Previously, work using this technique (PMID: 19164139) showed friulimicin B bound C55-P in an approximately equimolar (1:1) ratio, and did not inhibit downstream reactions (MurG, FemX, PBP2), suggesting specificity for C55-P of this compound. In this same study daptomycin at equimolar concentrations did not inhibit conversion of C55-P to lipid II. In the current work it is not until daptomycin:lipid molar ratios exceed 5-10:1 that we see an effect on lipid I synthesis and lipid II utilization. If daptomycin was forming a complex with undecaprenyl-intermediates, it is unclear why such a high molar ratio is needed to observe any effects? This finding seems to argue against a specific interaction, and rather describes a more general consequence of effects on the membrane. This would be consistent with what this group has previously shown, namely that daptomycin sequesters fluid lipids (including bactoprenol?) and displaces membrane associated proteins (PMID: 27791134), leading to inhibition of cell wall synthesis.

Indeed, our previous work (PMID: 19164139) on friulimicin showed that this lipopeptide binds to C55-P in stoichiometric manner and revealed specificity for C55-P, but not for C55-PP or lipid II.

However, despite the structural similarities at first glance, DAP differs largely from friulimicin.

Importantly, DAP forms oligomers consisting of 6 or more molecules (page 8, line 202 ff and page 10, line 244) as revealed by numerous studies and substantiated by the use of different experimental approaches and techniques (compare PMID: 22079564, 27237728, 21223947, 22387459, 28844744, 27288182 etc.). Thus, an almost complete inhibition observed at a 10:1 molar ratio (and an approx. 50% inhibition at 5:1) as observed in our current study is perfectly in line with oligomerization of DAP. We have highlighted this fact in the text and now directly link this information to the results. We pronounce this even more by discussing differences in the experimental set up compared to previous work using this technique (PMID: 19164139) and linking these results to friulimicin and amphomycin in the discussion (page 8, line 202 and 208; page 9, lines 236ff).

Indeed, we showed that our findings regarding fluid lipids (PMID: 27791134) are well in line with the findings of the current paper. We discussed this extensively (page 10-11).

Also see comment to point 4 regarding general effects on the membrane.

4. From the data presented in Figure 5, the authors suggest that the addition of oritavancin, bacitracin, and friulimicin block daptomycin binding to target membranes by blocking association with lipid intermediates. Did the authors exclude changes in membrane physicochemical properties, such as shifts in fluidity, as a potential source of decreased binding?

We did not specifically examine the mentioned physico-chemical membrane properties in this specific experiment (previous Fig. 5b; new Figure 4b). We have deliberately chosen three chemically diverse compounds that significantly differ in their hydrophobic moieties and affinity to the membrane. To limit changes in membrane properties potentially impacting on DAP binding, we extensively washed the planar bilayer to diminish unspecific (non-target mediated) binding of these antibiotics. Target recognition and specific binding are often reflected by a low dissociation rate as has been shown for oritavancin (PMID: 25403671) and result in strong interactions that are less affected by washing. Thus a reduction, especially almost equally and to the extent of DAP binding observed in the experiment, would not have been expected due to changes in membrane physicochemical properties.

5. Figure 3 demonstrates decreased daptomycin binding after pre-incubation with the lipid II- binding antibiotics teixobactin and oritavancin. I am unsure why they authors did not include the vancomycin data here as well. I would suggest incorporating the vancomycin data from Supplementary Figure 5 into the main Figure 3, using the same conditions and time points in both experiments.

We need to keep the figures separate, because the experiments presented focus on different aspects, demanding for different experimental conditions and approaches: in Figure 3 we address the effects on DAP binding and DAP-induced killing after pre-incubation with antibiotics that block target access and thus reduce DAP binding and killing. Therefore, incubation time needs to be short (2 min) followed by washing to avoid killing by the target blocking antibiotic. Instead, in Supplementary Figure 5 we wanted to increase lipid II target levels to enhance DAP binding. To achieve significantly increased levels of lipid II a prolonged incubation with VAN (30 min) is required.

We highlight this in the text by opposing the different approaches (page 6, line 152).

6. Further detail involving the methodology of Figure 5 would be helpful. Is the same location of fluorescence (over time, between samples) used? What percentage of the field of view over total area fluorescing? Were any efforts taken to determine the rate of diffusion of substances into the field of fluorescence over time? Lipid I might be useful to include as well. Further description of statistical analysis performed should be included in methods and appropriate statistical values should be marked in the Figure.

The imaged field of view with the extension $(32.5 \mu\text{m})^2$ corresponded approximately to $\frac{1}{4}$ of the total illuminated field. Thus, it can well be assumed that the observed field showed an isotropic fluorescence, and there was no influx of putatively unbleached and no efflux of putatively bleached molecules. This assumption is corroborated by an evaluation of the movie data, which did not show any spatial fluorescence gradient occurring over the field of view as a function of time. Still, in view of the above question/comment, we decided to evaluate only the first images of all movies in order to exclude any possible effect of im- or export of unbleached or bleached molecules, respectively. It can be seen that this approach did not alter the result of the experiments (please compare the new Fig. 4 and the former Fig. 5). In each data acquisition session, all samples were measured subsequently during one day in order to guarantee optimal comparability among the samples. From each sample at least 20 movies at different locations of the bilayer on each cover slip were acquired. Intensity values over a cropped region of 100×100 pixels corresponding to $161 \mu\text{m}^2$ within the first image of each movie were averaged in order to circumvent effects of possible photobleaching or molecular transport effect during movie data acquisition. The average background signal of the camera was subtracted and the intensity values were again averaged for each experimental condition separately. The observation field was defined by the camera position and was constant, however, small deviations in the very sensitive TIRF excitation beam path resulted in absolute intensity variations from day to day. As suggested by the reviewer, a student's t-test was performed to demonstrate the significance of the intensity differences between the respective samples (see new Fig. 4a, b). All experiments were repeated three times and yielded comparable results.

Including lipid I is pointless. In the "natural context" lipid I is located inside the cell and thus not accessible for DAP.

The text was modified accordingly in the „Methods“ section.

7. Supplementary Figure 6: Please specify the molar concentrations used in the experiment. If a 1:1 ratio of DAP:DOPG was used, and 1:1:1 of DAP:DOPG:C₅₅-P, then the second condition would be 1 part DAP to 2 parts lipid (DOPG, C₅₅-P), and may result in the observed decrease in fluorescence. Was a 1:0.5:0.5 mixture used?

We have now specified the concentrations used in the experiment and added the information to the figure labeling. In these experiments DOPG is always in excess (a condition also found in the living cell). The concentrations used correspond to molar ratios of 1:5:1 (DAP:DOPG:C₅₅-P). We assume that this reviewer refers to a possible charge effect resulting from the incorporation of C₅₅P to DAP:DOPG (1:5). To exclude unspecific charge effects the amount of PG was adjusted to equalize anionic charges of C₅₅P. In the new Figure 6b we have now included data showing molar ratios of 1:5, 1:6 and 1:7 (DAP:DOPG) compared to DAP:PG:C₅₅P at molar ratios of 1:5:1 and 1:5:2, respectively.

Minor Comments

1. The introduction has many overstated statements on the use of daptomycin and the clinical impact of infections treated with this antibiotics. Daptomycin is really not a last resort drug for staphylococci.

Disagree. There are no overstated statements. There are only two statements on the use of Daptomycin in the entire manuscript. The introduction doesn't even contain the term "last resort antibiotic", which has only been mentioned once in the abstract. Here we state: "Daptomycin is a last

resort antibiotic to treat severe infections with gram-positive pathogens, such as methicillin resistant *Staphylococcus aureus* (MRSA) and drug resistant enterococci.” There are hundreds of publications stating this antibiotic to be a compound of last resort for VRE and MRSA. In the introduction we now only state “Daptomycin (DAP) is a resistance breaking antibiotic with unprecedented biophysical properties and antibacterial activities.”

There is no doubt that DAP is resistance breaking in the context of its application.

In addition, see comment of Reviewer 1 stating that DAP is “one of the most important antibiotics in clinical use”.

2. The manuscript would benefit from revision of grammar as there are errors regarding comma usage, tense, sentence fragments and run-on sentences.

The manuscript has been revised by a native speaker and corrected.

3. The text frequently refers to daptomycin potentially binding to Lipid II (major comment above), however Figure 5 demonstrates potential strongest binding effect to C55P and then C55PP. Although the manuscript title reflects this, would alter wording throughout the manuscript to reflect this.

We will alter wording where relevant (e.g. page 13, line 330).

4. In Figure 1, the concentrations of daptomycin differ from what is written in the corresponding section in the Materials and Methods.

Has been corrected.

5. Supplementary Figure 3 is superfluous, a citation directing the reader to its source (PMID: 23215859) would be sufficient.

This is a simplified version of the figure published in PMID: 23215859. We suggest to keep this figure in the supplement, since we think that it is helpful for a broader readership. Also, see comment of reviewer 1, requesting more information on MraY, which is facilitated by this figure.

6. Would include fluorescence intensity in Figure 3b.

Figure 3b has been optimized. See also comment 8 of reviewer 1.

7. In Supplementary Figure 4, I believe the middle column of images should be labeled “FtsW-GFP” not “DAP-FL.”

Correct, thank you. The figure has been changed accordingly. In the course of the revision we decided to combine Figure 4 and Suppl. Figure 4 (new Suppl. Figure 4).

REVIEWERS' COMMENTS:

Reviewer #1 (Remarks to the Author):

All my concerns have been appropriately addressed. Andreas Peschel

Reviewer #2 (Remarks to the Author):

I appreciate the extensive revisions of the initial manuscript, where the authors have expanded and clarified important aspects of the work. I am satisfied with the authors' response. The following comments are for authors' consideration:

1- I wished to point out with the prior cited references that in *S. aureus* the LiaRS homologue is *VraSR*, and that activity of this system is tied to daptomycin resistance. I was unsure why they chose to examine the LiaRS expression in *Bacillus*, while many of the experiments were focused on *S. aureus*, given the prior heterogeneity of model organisms. Having said that, I believe this is a minor point and clarifications in the revised manuscript are useful.

2- The biphasic binding described by the authors is an interesting phenomenon and I appreciate that they decided to highlight this phenomenon, which is novel. I agree that the model is in line with the author's prior published data, and the potential lipid extraction effect daptomycin has shown on giant unilaminar vesicles (GUVs). However, since septal binding is associated with bacterial cell death, partially due to sequestration of undecaprenyl-intermediates, it is unclear whether these observations are contributing to, or a consequence of, the proposed mechanism of action.

3- Regarding lipid I, I do not view this as "pointless", since daptomycin has been implicated in translocation to the inner leaflet of the bilayer and formation of an oligomeric complex across the membrane (PMID: 24616102). In fact, that study suggested that cardiolipin restricted the translocation of daptomycin from outer to inner leaflet, and changes in cardiolipin content of bacterial membranes have been implicated in resistance to daptomycin. One could speculate that oligomerization on both sides of the membrane (rather than only the external leaflet as depicted in figure 7), be important (possibly necessary) for the action of daptomycin. I do not suggest that the authors perform more experiments since they would be beyond the scope of the current manuscript, but would be something for them to consider.

4- A minor point, I still do not understand the term used for the authors as "resistance breaking" when referring to DAP. Clinically this drug has many limitations that have become clear with the extensive use of the drug. Again, I respectfully disagree with the use of the term "last resort". Definitely, not for *S. aureus*

5- I really appreciate the addition of figure 7 depicting the model of DAP mechanism of action.

Response to REVIEWERS' COMMENTS:

REVIEWERS' COMMENTS:

Reviewer #1 (Remarks to the Author):

All my concerns have been appropriately addressed.

Reviewer #2 (Remarks to the Author):

I appreciate the extensive revisions of the initial manuscript, where the authors have expanded and clarified important aspects of the work. I am satisfied with the authors' response. The following comments are for authors' consideration:

1- I wished to point out with the prior cited references that in *S. aureus* the LiaRS homologue is *VraSR*, and that activity of this system is tied to daptomycin resistance. I was unsure why they chose to examine the LiaRS expression in *Bacillus*, while many of the experiments were focused on *S. aureus*, given the prior heterogeneity of model organisms. Having said that, I believe this is a minor point and clarifications in the revised manuscript are useful.

2- The biphasic binding described by the authors is an interesting phenomenon and I appreciate that they decided to highlight this phenomenon, which is novel. I agree that the model is in line with the author's prior published data, and the potential lipid extraction effect daptomycin has shown on giant unilamellar vesicles (GUVs). However, since septal binding is associated with bacterial cell death, partially due to sequestration of undecaprenyl-intermediates, it is unclear whether these observations are contributing to, or a consequence of, the proposed mechanism of action.

3- Regarding lipid I, I do not view this as "pointless", since daptomycin has been implicated in translocation to the inner leaflet of the bilayer and formation of an oligomeric complex across the membrane (PMID: 24616102). In fact, that study suggested that cardiolipin restricted the translocation of daptomycin from outer to inner leaflet, and changes in cardiolipin content of bacterial membranes have been implicated in resistance to daptomycin. One could speculate that oligomerization on both sides of the membrane (rather than only the external leaflet as depicted in figure 7), be important (possibly necessary) for the action of daptomycin. I do not suggest that the authors perform more experiments since they would be beyond the scope of the current manuscript, but would be something for them to consider.

We thank this reviewer for the comments and the valuable input which we will definitely consider in future studies as suggested.

4- A minor point, I still do not understand the term used for the authors as "resistance breaking" when referring to DAP. Clinically this drug has many limitations that have become clear with the extensive use of the drug. Again, I respectfully disagree with the use of the term "last resort". Definitely, not for *S. aureus*

We have deleted the terms “resistance breaking” and “last resort”.

5- I really appreciate the addition of figure 7 depicting the model of DAP mechanism of action.

We thank both reviewers for their time and input which helped to improve the manuscript a lot.